# Effects of Soybean Trypsin Inhibitor on Pancreatic Oxidative Damage of Mice at Different Growth Periods

**DOI:** 10.3390/foods12081691

**Published:** 2023-04-18

**Authors:** Chunmei Gu, Qiuping Yang, Shujun Li, Linlin Zhao, Bo Lyu, Yingnan Wang, Hansong Yu

**Affiliations:** 1College of Food Science and Engineering, Jilin Agricultural University, Changchun 130118, China; jjnong2008@126.com (C.G.); zhaoll89@163.com (L.Z.); michael_lvbo@163.com (B.L.); 2Division of Soybean Processing, Soybean Research & Development Center, Chinese Agricultural Research System, Changchun 130000, China; virginiay@163.com; 3Heilongjiang Green Food Science Research Institute, Northeast Agricultural University, Harbin 150030, China; 4Department of Agriculture and Resources Environment, Qinghai Higher Vocational and Technical College, Haidong 810799, China; lishujun1026@163.com; 5College of Tourism and Culinary Science, College of Food Science and Engineering, Yangzhou University, Yangzhou 225127, China

**Keywords:** soybean trypsin inhibitor, pancreas, oxidative damage, genetic expression

## Abstract

The bioactive components in soybeans have significant physiological functions. However, the intake of soybean trypsin inhibitor (STI) may cause metabolic disorders. To investigate the effect of STI intake on pancreatic injury and its mechanism of action, a five-week animal experiment was conducted, meanwhile, a weekly monitor on the degree of oxidation and antioxidant indexes in the serum and pancreas of the animals was carried out. The results showed that the intake of STI had irreversible damage to the pancreas, according to the analysis of the histological section. Malondialdehyde (MDA) in the pancreatic mitochondria of Group STI increased significantly and reached a maximum (15.7 nmol/mg prot) in the third week. Meanwhile, the antioxidant enzymes superoxide dismutase (SOD), glutathione peroxidase (GSH-Px), trypsin (TPS), and somatostatin (SST) were decreased and reached minimum values (10 U/mg prot, 87 U/mg prot, 2.1 U/mg prot, 10 pg/mg prot) compared with the Group Control. The RT-PCR results of the expression of SOD, GSH-Px, TPS, and SST genes were consistent with the above. This study demonstrates that STI causes oxidative structural damage and pancreatic dysfunction by inducing oxidative stress in the pancreas, which could increase with time.

## 1. Introduction

The anti-nutritional factors in soybeans can inhibit the growth of animals by interfering with digestion, absorption, and utilization of nutrients [1,2], which limits the application of soybeans in foodstuff and animal feeding because of the necessity of heat treatment and temperature control in this process. As one of the main soybean anti-nutritional factors, soybean trypsin inhibitors (STI) is a polypeptide composed of 72–197 amino acid residues [2], which may cause some physiological reactions, such as pancreatic hypertrophy and even pancreatic cancer [3,4]. The ingestion of diets containing STI by animals can result in the formation of a complex between trypsin and chymotrypsin with the STI in the intestinal tract. This complex can then be excreted, leading to a reduction in enzyme activity. Consequently, the pancreas may attempt to compensate for the reduced enzyme activity by increasing its secretion and synthesis of trypsin [4,5]. In addition, the synthesis of DNA, mRNA, and enzymes require ATP for purine and pyrimidine synthesis, as well as the activation of amino acids. However, the production of ATP also generates free radicals. Similarly, a large amount of ATP is required by the STI-stimulated pancreas for trypsin synthesis. As a result, consuming STIs may also result in elevated levels of oxygen free radicals.

The excess free radicals, regardless of whether they are generated by the mitochondrial respiratory chain or NAD(p)H, can cause oxidative stress, which has a direct impact on cells. This can result in cell damage and eventually various diseases [6,7,8,9], such as cardiovascular diseases, cancer, neurological disorders, diabetes, ischemia/reperfusion, and aging [10,11,12,13,14,15,16,17], which is also one of the risks of STI intake. In vivo, Vitamin C (VC) can react with oxygen free radicals through redox reactions, thereby neutralizing them and safeguarding the body against their damaging effects. Therefore, VC is widely recognized as an antioxidant that can effectively prevent oxidative damage to the pancreas.

While STIs were initially developed as drugs, their potential side effects on the body from daily consumption were not thoroughly understood. This study was conducted to examine the impact of STI consumption on the pancreas, including its effects on pancreatic structure, function, and gene expression in mice with varying growth cycles. Additionally, it investigated the effects of STI on free radical levels during different stages of growth and the degree of oxidative damage to the mouse pancreas, which the potential mechanism was studied, and explored whether the intake of antioxidants can reduce pancreatic oxidative damage. This work offered a theoretical foundation for identifying endogenous strategies to prevent pancreatic injury resulting from the intake of STIs.

## 2. Material and Methods

### 2.1. Main Reagents and Reasons for Selection

Soybean trypsin inhibitors (STI), the most common enzyme inhibitors in soybeans, are the most significantly damaging to the pancreas by their ingestion. Additionally, as a strong antioxidant, Vitamin C (VC) has a theoretical potential to mitigate oxidative damage to the pancreas. Therefore, both were chosen to conduct this study.

STI, in which the activity was identified as 4600 BAEE U/mg, was provided by the College of Food Science and Engineering, Jilin Agricultural University. Vitamin C (Ascorbic acid, A8100) was provided by Beijing Solarbio Science & Technology Co., Ltd. (Beijing, China).

### 2.2. Animals and Diets

Four- to six-week-old KM male mice were purchased from Changchun Yisi Experimental Animal Technology Co., Ltd. (Changchun, China). All animals were housed under a controlled condition in individual cages at 23 ± 2 °C and 50 ± 10% relative humidity with a 12 h light/dark cycle in a specific pathogen-free environment and were allowed free access to food and water.

After one week of acclimatization, the mice were divided into three groups randomly: the control group (control diet, *n* = 10), Group STI (control diet containing 2.0 mg/g STI, *n* = 10), Group STI + VC (STI diet supplemented with 1500 mg/kg VC, *n* = 10). A total of five intergroup parallels were set up for each group for the weekly sacrifice of animals (Total: 150 mice). All the mice were sacrificed after a 5-week feeding. All the animal experiments were approved by the Institutional Animal Care and Use Committees of Jilin Agricultural University (Protocol code No.20130530001, May 2013), following the National Research Council Guidelines. The composition of the control diet is shown in Table 1.

### 2.3. Sample Preparation

At the end of each week, the mice that were to be sacrificed were fasted for 12 h but had free access to deionized water. Blood was obtained from the eyeballs of mice and centrifuged at 4000× *g* for 3 min at 4 °C using a high-speed desktop refrigerated centrifuge (TGL-16G, Shanghai Anting Scientific Instrument Factory, Shanghai, China), and serum was separated and stored at −20 °C for a maximum of 16 weeks. Then mice were sacrificed by decapitation, and the pancreas was quickly removed, gently rinsed in ice-cold PBS (Wuhan Punuosai Life Technology Co., Ltd., Wuhan, China), and cut into 50–100 mg/100 g body weight, frozen in liquid nitrogen and stored at −80 °C for the follow-up experiments. After thawing at 4 °C, tissue samples were homogenized using a MagNALyser instrument (Roche Diagnostics, Mannheim, Germany) at 4000× *g* for 50 s twice, and then diluted with 9 volumes ice-cold 0.9% NaCl solution, then centrifuged at 4000× *g* for 15 min at 4 °C. Functional compounds, oxidative, and antioxidant activity were analyzed using the supernatants collected from the samples. All operations were done at 4 °C. Protein content was measured using the method of Lowry (Lowry et al., 1951) with bovine serum albumin as a standard, assuming it to be 100% pure. Protein content was expressed as BSA equivalents.

### 2.4. Analytical Methods

#### 2.4.1. Determination of Oxidation and Antioxidant Parameters

The malondialdehyde (MDA) content was measured using the TBA reaction method of Koca et al. [18]. After the preparation, according to the kit used (A003-1-1, Nanjing Jiancheng Bioengineering Institute, Nanjing, China), samples were incubated in a 95 °C water bath for 40 min and centrifuged at 3500× *g* for 10 min. The absorbance of the supernatant was measured with a double beam UV-Vis spectrophotometer (T6 New Century, Beijing General Instrument Co., Ltd., Beijing, China) at 532 nm, which was attributed so that the MDA could be condensed with TBA to form a red product with the maximum absorption peak at 532 nm. Results were expressed as units nmol/mg of protein for pancreas samples, and nmol/mL for serum samples. The activity of superoxide dismutase (SOD) was measured using the method of Beauchamp & Fridovich [19]. The samples were prepared according to the instructions of the kit (A001-1-1, Nanjing Jiancheng Bioengineering Institute, Nanjing, China) and then placed in a water bath at 37 °C for 40 min. In the process, O_2_^−^∙ was produced by the reaction system of xanthine and xanthine oxidase and could oxidize hydroxylamine to form nitrite. Under the action of a chromogenic agent, nitrite could appear purple-red and have a maximum absorption peak at 550 nm, and results were expressed as units U/mg of protein for pancreas samples and U/mL for serum samples. Glutathione peroxidase (GSH-Px) activity was measured according to the method of Sabuncu et al. [20], and results were shown as units U/mg of protein for pancreas samples and U/mL for serum samples.

#### 2.4.2. Determination of Trypsin Activity and Hormone Levels

The trypsin (TPS) activities in serum and tissue samples were assayed at 410 nm using the substrate N-benzoyl-DL-arginine p-nitroaniline hydrochloride (BAPNA) (A080-2-2, Nanjing Jiancheng Bioengineering Institute, Nanjing, Jiangsu, China) according to the manufacturer’s instructions [21]. TPS could react with BAPNA to release p-nitroaniline with a maximum absorption peak at 410 nm. Somatostatin (SST) of serum samples and pancreas tissue samples were measured using a radio-immunoassay method (bs-1132R, Anti-Somatostatin/GR, Beijing Huaying Bioengineering Institute, Beijing, China) [22]. The specific antibodies were bound to a solid-phase carrier to form a solid-phase antibody and then combined with the corresponding antigen in the samples to form an immune complex. Then the enzyme-labeled antibody was bound to the antigen in the immune complex to form an enzyme-labeled antibody-antigen-solid phase antibody complex. This complex could be colored by adding a substrate and was with a maximum absorption peak of 450 nm.

#### 2.4.3. Transmission Electron Microscopy (TEM) of the Pancreas

Pancreas tissues that had been fixed with 2.5% glutaraldehyde (Sigma Aldrich Co., St. Louis, MO, USA) were removed from the glutaraldehyde and treated as follows: Samples were post-fixed with 1% osmic acid at 4 °C for 2 h, dehydrated with gradient concentrations of acetone (once with 50%, 70%, and 90% acetone and three times with 100% acetone for 15 min each), and embedded in Epon812 (Beijing Zhongjing Keyi Technology Co., Ltd., Beijing, China) at room temperature (22–25 °C) overnight. The samples were sliced into 5 μm sections with a rotary microtome (Leica Microsystems (Shanghai) Trading Co., Ltd., Shanghai, China), counterstained with 2% (*w*/*v*) uranyl acetate and lead citrate (SPI-CHEM, West Chester, PA, USA), and then observed using TEM at 80 kV and a magnification of 12,000× (Hitachi High-tech (Shanghai) International Trade Co., Ltd., Shanghai, China).

#### 2.4.4. RNA Extraction and Real-Time PCR

Total RNA was obtained from the pancreas samples using an RNeasy Minikit (Qiagen, Hilden, Germany). The pancreas sample was fully ground, then 20 mg was taken out, 350 μL of lysate was added, and RNA extraction was performed according to the instructions of the kit and resuspended in 50 μL RNase-free water (included with the kit). Synthesis of cDNA was primed by oligo d (T) using a PrimeScript RT Enzyme (Takara, Beijing, China) according to the Power SYBR Green PCR Master Mix (Life Technologies, Beijing, China) instructions. The primers were synthesized by Shanghai Biological Engineering Design Services Ltd. (Shanghai, China). The reaction system for the synthesis of the first strand of cDNA was 5× PrimeScript buffer 4 µL, Template RNA 1 µL, PrimeScript RT Enzyme Mix I 1 µL, Oligo dT primer 1 µL, RNase-free dH_2_O 12 µL, and Random 6 mers 1 µL, total reaction volume 20 µL. Before adding the reverse transcriptase, the mixed solution was first dried at 700 °C for 3 min. After taking it out, the temperature inside and outside the tube was the same, then reverse transcriptase was added, and the 37 °C water bath was used for 15 min. Immediately after it taking out, it was put in a dry bath at 85 °C for 5 s to obtain cDNA solution and stored at −80 °C until used. The reactions were done in a thermocycler StepOne™ Real-Time PCR System. The master mix prepared for analysis of each gene was composed of 0.5 μL forward primer, 0.5 μL reverse primer, 10 μL of 2× SYBR premix, and 1 μL cDNA in a total volume of 20 μL. Each sample was analyzed in triplicate. RT-PCR was done using the following conditions: reverse transcription at 50 °C for 2 min, PCR activation at 95 °C for 10 min followed by 40 cycles of denaturation at 95 °C for 15 s, annealing at 60 °C for 1 min, and a final extension at 72 °C for 10 min. β-Actin was the internal reference gene. Sequences of primers are shown in Table 2 (Designed by Shanghai Biotech Biotechnology Co., Ltd., Shanghai, China).

### 2.5. Statistical Analysis

Data were reported as mean ± SD, *n* = 10 (per week). Differences between mean values were determined using a one-way ANOVA followed by comparisons using the Newman-Keuls multiple range test. Differences with *p* < 0.05 were considered significant. Statistical analyses were done using the Statistical Program for the Social Sciences, SPSS software (Version 22.0, SPSS Inc., Chicago, IL, USA).

## 3. Results

### 3.1. The Pancreas Index, Oxidative and Antioxidant Parameters in Serum and Pancreas of Mice

As shown in Table 3, the pancreas weight of the STI group and STI + VC group showed a trend of first increasing and then decreasing, as compared to the control group. However, due to significant individual differences, there was no significant difference except for the first week.

The levels of MDA in the serum and pancreas increased and subsequently decreased in both the STI and STI + VC groups, with the maximum levels observed during the 3rd week. (Table 4). Mice fed STI and STI + VC showed a significant increase (*p* < 0.05) in the serum MDA content compared with the control group in the first 4 weeks and showed no differences in the 5th week. The addition of VC caused a significant decrease (*p* < 0.05) of the MDA level in the serum than mice in the STI group, except in the 2nd and 5th weeks.

In the pancreas, it showed a significant increase (*p* < 0.05) in MDA content in the STI group and STI + VC group in comparison with those of the control mice, except in the 5th week. MDA content of the STI + VC group was significantly decreased (*p* < 0.05) when compared with those in the STI group in the 3rd, 4th, and 5th weeks.

As shown in Table 5 and Table 6, as the feeding periods increased, SOD and GSH-Px activities of the three groups in both the serum and pancreas decreased and then increased and, in the 3rd week, reached a minimum. Compared with the control animals during the whole time, mice fed STI and STI + VC showed a significant decrease (*p* < 0.05) in SOD and GSH-Px activities. Meanwhile, the activities of SOD and GSH-Px in the STI + VC group were significantly higher (*p* < 0.05) than those in the STI group for the majority of the time. Therefore, STI can significantly reduce the level of SOD and GSH-Px in mice, and STI + VC can slightly increase the level of SOD and GSH-Px compared with the STI group.

### 3.2. Trypsin Activity and Hormone Levels

The activities of TPS are summarized in Table 7. As the time increased, the TPS activity of the STI group and STI + VC group in both the serum and pancreas decreased initially and then increased and, in 3rd week, reached a maximum. Mice fed STI and STI + VC showed significant decreases in TPS activity when compared with the control animals during the whole period. Whereas the TPS activity in the STI + VC group was significantly higher (*p* < 0.05) than those in the STI group in the majority of the whole periods.

The contents of SST are summarized in Table 8. With the increase in feeding periods, the SST contents of three groups in both the serum and pancreas were decreased and then increased and, in the 3rd week, reached a minimum. Mice fed STI and STI + VC showed a significant decrease (*p* < 0.05) in SST content when compared with the control animals during the whole period except for that in the serum for the 2nd and 3rd week and the pancreas for the 3rd week. Meanwhile, the SST content in the STI + VC group was higher (*p* < 0.05) than that in the STI group for the majority of the whole period.

### 3.3. Analysis of Relative Gene Expression

Five relative genes: GSH-Px, SOD, TPS, SST, and SSTR5, were analyzed using RT-PCR (Figure 1). The transcription levels of GSH-Px, SOD, TPS, SST, and SSTR5 genes in the three groups exhibited a decreasing trend during the initial 3 weeks, reaching a minimum in the 3rd week, followed by a subsequent increase during the next 2 weeks. During the whole experimental period, there was a significant decrease (*p* < 0.05) in the transcription levels of GSH-Px, SOD, TPS, and SST genes in the mice of the STI group compared to the control group. Mice in the STI + VC group exhibited significantly higher transcription levels (*p* < 0.05) of SOD, GSH Px, TPS, and SSTR5 genes, as compared to those in the STI group, although still lower than the control group.

### 3.4. TEM of Pancreas Tissue

Figure 2 shows the ultrastructure changes in the pancreas of three groups of mice. The analysis of the pancreas showed no pathological alterations in control mice. The nuclear envelope, nucleus, mitochondria, and endoplasmic reticulum were normal. Zymogen granules were diffusely distributed in the cytoplasm (Figure 2a). Since the oxidative damage was less pronounced in the STI group during the first week, the electron micrographs exhibited similarity to those of the control group. Observable damage appeared in the 3rd week, with micrographs of the pancreas of the STI group displaying mitochondrial vacuolization, swelling, and dilatation of the endoplasmic reticulum. The zymogen granules of STI-diet mice were significantly fewer than that of control mice (Figure 2b,d,f,h,j). The protective effect of VC in mice was evident in the form of a moderate increase in zymogen granules, as well as normal mitochondria and endoplasmic reticulum. Mild swelling of the endoplasmic reticulum and mitochondria were also observed, indicating the efficacy of VC in preventing damage caused by STI, as compared to the STI group (Figure 2c,e,g,i,k).

## 4. Discussion

The lipid peroxidation process is initiated by reactive oxygen species in the phospholipids present in biofilms, triggering a free radical chain reaction [23]. As MDA is the result of lipid peroxidation, its concentration can be used as an indicator to measure the extent of lipid peroxidation. The results indicated a significant rise in the MDA level in the serum and pancreas of the STI group when compared to the control group. These results indicated that STI induced a significant increase in lipid peroxidation, indirectly indicating a rise in the levels of free radicals in the pancreas of mice. As the feeding time increased, it was observed that MDA levels in the STI group increased significantly, reaching their peak in the 3rd week. Studies have shown that in the myocardial ischemia and hyperlipidemia animal model [24,25], the level of MDA will rise first and then decline, which was consistent with the findings of this study. The results indicated that MDA levels exhibit periodic fluctuations and are influenced by feeding time, consistent with the pattern observed for free radicals in the experiment. This further demonstrated that the growth cycle of mice influences the level of free radicals affected by STI, with the highest level reached in the 3rd week. TEM analysis of the pancreas also revealed that oxidative damage to the pancreas was most severe in the 3rd week.

As a free radical scavenging system, superoxide dismutase (SOD) and glutathione peroxidase (GSH) exist in all oxygen-metabolizing cells, which can prevent free radical damage to cells and provide an oxidative membrane component of repair mechanism [14], reflecting the capacity of the non-enzymatic antioxidant defense system. Therefore, the aforementioned parameters were utilized to evaluate alterations in the antioxidant status of the pancreas. The findings showed a decrease in the antioxidant indices of both the serum and pancreas in the STI group, which was caused by the STI increasing the level of oxygen free radicals. To maintain the balance between oxidation and antioxidants, the body requires a significant amount of antioxidants. However, this high demand results in damage to the antioxidant defense system, leading to a weakened antioxidant capacity within the body. The activity of SOD in both the serum and pancreas displayed a pattern of initially decreasing, followed by increasing and subsequently reaching the lowest level during the third week. However, the activity of GSH-Px showed an opposite trend. RT-PCR results indicated the down-regulation of transcription levels of SOD and GSH-Px genes in the STI group when compared to the control group. This is due to the STI leading to an increase in free radicals.

The expression of SOD and GSH-Px genes can be inhibited by free radicals, leading to a significant reduction in the activity of SOD and GSH-Px [26]. These results provide more insight into the alteration in antioxidant capacity following oxidative stress, which could be associated with the level of oxygen-free radicals and the developmental stage of the organism.

It was observed that the TPS of the STI group and STI + VC group exhibited a decreasing trend, followed by an increasing trend as the feeding time increased, which reached their minimum value in the third week. TPS, produced in the pancreas, is an endopeptidase that binds to trypsin inhibitors to create a complex of enzymes and inhibitors. During the last two weeks, as oxidative damage decreased, the activity of TPS increased, which led to the formation of these complexes that can be excreted in feces, causing a decrease in trypsin levels.

SST is a natural, ubiquitous neurohormone found in the central nervous system and most major peripheral organs, including the salivary glands, stomach, pancreas, and intestine [27]. It is believed that this peptide has negative effects on various physiological functions. The action of SST is mediated by the 5-somatostatin receptor subtype, known as SSTR1-5. The levels of SSTR5 in the pancreas were higher than those of the other four receptors. SST inhibits the secretion of both insulin and glucagon, which is mediated through distinct SST receptors [28]. In addition, SST has the ability to not only directly impact islet secretion function but also modulate islet function by influencing the responsiveness of islet cells to physiological or pharmacological stimuli [29,30]. Recent studies have shown that various factors that stimulate the release of SST from cells can also trigger the production of intracellular ROS. Wenger’s study concluded that the SST analog octreotide impacted oxygen-free radical metabolism by reducing liver re LPO, increasing GSH-Px and SOD activity [31]. The results of this experiment showed that the SST content in the STI group first increased, then decreased, and peaked during the 3rd week. However, RT-PCR results indicated that the transcription levels of SST and SSTR5 genes in the STI group exhibited an inverse correlation. This may be due to the inhibition of the expression of the SST gene and the SSTR5 gene during transcription due to the production of free radicals. The possible reason for the increase in SST content is the autocrine and paracrine factors of SST. In other experimental conditions, SST has demonstrated its effectiveness as an inhibitor of insulin secretion [32]. While promoting SST production, insulin secretion is suppressed. However, in our study, we observed an opposite trend between insulin and SST levels.

VC is involved in metabolic processes in the body and acts as a potent antioxidant by directly or indirectly neutralizing free radicals to prevent cellular damage and immune system disorders [33]. The results of this study suggest that VC may alleviate the oxidative stress caused by STI, thus mitigating the oxidative damage caused by reactive oxygen species.

## 5. Conclusions

The current research indicates that STI exhibits the harmful effect of inducing oxidative stress by increasing the formation of lipid peroxidation and overall impairing enzymatic and non-enzymatic antioxidant defenses in the STI diet-fed mice, which cause structural damage and secretory dysfunction of the pancreas. Moreover, the RT-PCR results for the expression of SOD, GSH-Px, TPS, SST, and SSTR5 genes further demonstrated the above results. In addition, these harmful effects are periodic. Meanwhile, the inference that free radicals generated by STI intake where the main contributor to pancreatic injury was confirmed by the improvement in pancreatic injury after VC intake. This suggested that adding VC, especially soy protein products, to ingredients containing STIs, whether food or feed, is a good way to mitigate the damage caused by STIs to the pancreas or that there are practical implications of consuming soy protein (containing STI) along with Vc.

## Figures and Tables

**Figure 1 foods-12-01691-f001:**
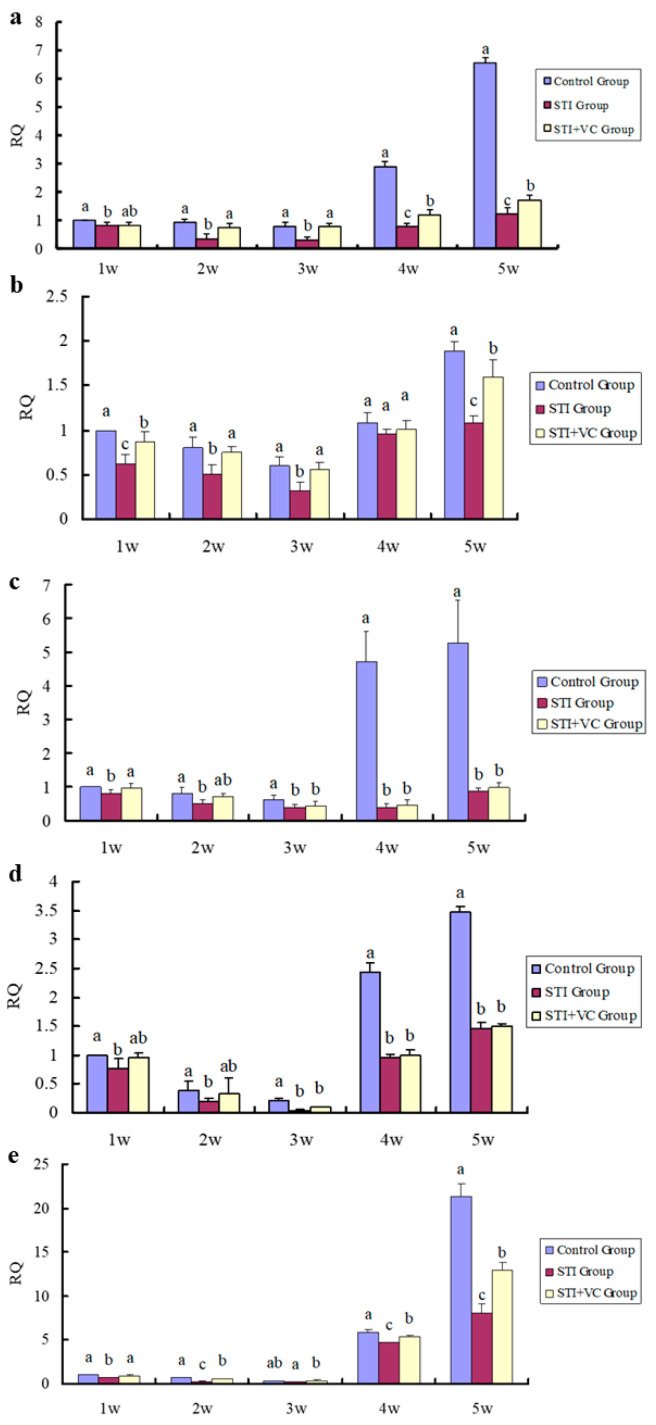
RT-PCR gene expression analysis of relative gene expression. (**a**) SOD; (**b**) GSH-Px; (**c**) TPS; (**d**) SST; (**e**) SSTR5; Different letters indicate significant differences between the groups (*p* < 0.05, *n* = 10).

**Figure 2 foods-12-01691-f002:**
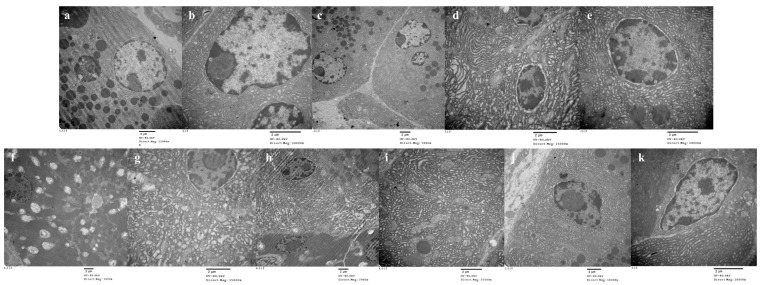
Electron micrographs of mice pancreas. (**a**) Pancreatic ultrastructure of the control group; (**b**,**d**,**f**,**h**,**j**) Ultrastructure of the pancreas in STI group at 1–5 weeks; (**c**,**e**,**g**,**i**,**k**) Ultrastructure of the pancreas in STI + VC group at 1–5 weeks.

**Table 1 foods-12-01691-t001:** Composition of the control diet ^a^.

	Ingredient	Diet (g/kg)
Protein	Casein	200
Carbohydrates	Corn starch	660
Fat	Soybean oil (without STI)	50
Fiber	Cellulose powder	30
Others	Mineral mixture ^b^	50
	Vitamin mixture ^c^	10

Note: The control group chow diet in pellet form (standard chow diet) was provided by the Changchun Yisi Experimental Animal Technology Co., Ltd., Jilin, China. The chow diets of the STI group and the STI + VC group were prepared by adding STI (2.0 mg/g) and VC (1500 mg/kg) into the standard chow diet, respectively, and then pelleted (prepared by the Agricultural Products Processing and Storage Engineering Laboratory of Jilin Agricultural University). ^a^ The diets were semi-purified (added mineral and vitamin complex additives), and isoenergetic was calculated according to China national standard GB14924.1-2001 (group I 16.3 MJ/kg, group II 15.7 MJ/kg, group III 15.5 MJ/kg). ^b^ The mineral mixture provides the following amounts (g/kg diet): Ca, 4; K, 2.4; Na, 1.6; Mg, 0.4; Fe, 0.12; trace elements (mg/kg diet): Mn, 32; Cu, 5; Zn, 18; Co, 0.04; I, 0.02. ^c^ The vitamin mixture provides the following amounts (mg/kg diet): retinol, 12; cholecalciferol, 0.125; thiamin, 40; riboflavin, 30; pantothenic acid, 140; pyridoxine, 20; inositol, 300; cyanocobalamine, 0.1; ascorbic acid, 1600; (dL) α-tocopherol, 340; menadione, 80; nicotinic acid, 200; para-aminobenzoic acid, 100; folic acid, 10; biotin, 0.6; choline, 2720.

**Table 2 foods-12-01691-t002:** The sequence of primers designed for the RT-PCR studies.

Gene Product	Primer Sequence	T (°C)	PCR (bp)
Glutathione Peroxidase (GSH-Px)	5′-TGGCATTGGCTTGGTGATTACTGG-3′(F)	59	150
	5′-GGTGGAAAGGCATCGGGAATGG-3′(R)	60	
Superoxide Dismutase (SOD)	5′-CCTTGTGACTGGCATCCCTTAGC-3′(F)	58	105
	5′-AGGCAGACTGTTAGATGGCTTGTTC-3′(R)	59	
Somatostatin (SST)	5′- CCTCTCCCATTCCTCCCTTTTGTTC-3′(F)	59	108
	5′-GGGCATCATTCTCTGTCTGGTTGG-3′(R)	58	
Somatostatin Receptor 5 (SSTR5)	5′-CGTCTGTGCTGGGCTTCTTTGG-3′(F)	60	136
	5′-ATGCGAGTCACCTTGCGTTCTG-3′(R)	58	
Trypsin (TPS)	5′-TCCTCATCTCTACCCACAACATTGC-3′(F)	60	96
	5′-CACTTCCGAACCATAACCGTAGGC-3′(R)	58	

**Table 3 foods-12-01691-t003:** Effect of different diets on the ratio of Pancreas/body weight in mice.

Treatment	1 Wk	2 Wk	3 Wk	4 Wk	5 Wk
Control	0.09 ± 0.01	0.09 ± 0.01	0.09 ± 0.02	0.09 ± 0.01	0.08 ± 0.01
STI	0.11 ± 0.01 *	0.10 ± 0.02	0.12 ± 0.02	0.11 ± 0.02	0.10 ± 0.01
STI + VC	0.10 ± 0.02	0.10 ± 0.01	0.11 ± 0.01	0.09 ± 0.02	0.09 ± 0.02

*: Represents the significant difference compared to the control group (*p* < 0.05, *n* = 10).

**Table 4 foods-12-01691-t004:** MDA content in serum and pancreas.

	Treatment	1 Wk	2 Wk	3 Wk	4 Wk	5 Wk
Serum (nmol/mL)	Control	2.5 ± 0.3	3.4 ± 0.2	3.8 ± 0.2	4.1 ± 0.2	4.5 ± 0.3
	STI	4.4 ± 0.2 *	4.7 ± 0.1 *	7.6 ± 0.2 *	5.6 ± 0.1 *	4.9 ± 0.1
	STI + VC	3.2 ± 0.1 ^#^	4.6 ± 0.4 *	5.9 ± 0.2 ^#^	4.6 ± 0.3 *	4.8 ± 0.1
Pancreas (nmol/mg prot)	Control	4.2 ± 0.2	4.9 ± 0.3	5.5 ± 0.3	7.4 ± 0.2	8.1 ± 0.3
	STI	6.3 ± 0.1 *	8.3 ± 0.2 *	15.7 ± 0.4 *	11.7 ± 0.4 *	9.6 ± 0.3 *
	STI + VC	6.1 ± 0.1 *	8.1 ± 0.2 *	13.7 ± 0.3 ^#^	9.6 ± 0.4 ^#^	8.6 ± 0.3

*: Represents the significant difference compared to the control group (*p* < 0.05, *n* = 10). ^#^: Represents the significant difference compared to the STI group (*p* < 0.05, *n* = 10).

**Table 5 foods-12-01691-t005:** SOD activity in serum and pancreas.

	Treatment	1 Wk	2 Wk	3 Wk	4 Wk	5 Wk
Serum (U/mL)	Control	174 ± 3.2	162 ± 4.5	135 ± 3.5	147 ± 3.8	117 ± 3.5
	STI	150 ± 2.8 *	120 ± 10 *	62 ± 1.8 *	83 ± 0.4 *	92 ± 1.9 *
	STI + VC	162 ± 1.1 ^#^	133 ± 3.0 *	77 ± 1.7 ^#^	94 ± 2.2 ^#^	106 ± 4.0 ^#^
Pancreas (U/mg prot)	Control	81 ± 4.7	68 ± 1.3	50 ± 1.0	39 ± 1.5	22 ± 1.8
	STI	49 ± 3.5 *	38 ± 2.8 *	10 ± 1.7 *	18 ± 0.3 *	16 ± 0.2 *
	STI + VC	55 ± 3.5 *	47 ± 1.3 ^#^	18 ± 0.4 ^#^	23 ± 1.2 ^#^	18 ± 1.7 ^#^

*: Represents the significant difference compared to the control group (*p* < 0.05, *n* = 10). ^#^: Represents the significant difference compared to the STI group (*p* < 0.05, *n* = 10).

**Table 6 foods-12-01691-t006:** GSH-Px activity in the serum and pancreas.

	Treatment	1 Wk	2 Wk	3 Wk	4 Wk	5 Wk
Serum (U/mL)	Control	693 ± 33	359 ± 28	260 ± 10	496 ± 35	1380 ± 30
	STI	481 ± 32 *	315 ± 57 *	165 ± 31 *	410 ± 10 *	577 ± 45 *
	STI + VC	540 ± 28 ^#^	345 ± 41 ^#^	240 ± 10 ^#^	456 ± 44 ^#^	640 ± 30 *
Pancreas (U/mg prot)	Control	287 ± 12	190 ± 10	180 ± 10	240 ± 10	384 ± 1.7
	STI	134 ± 14 *	96 ± 4.5 *	87 ± 4.8 *	108 ± 5.5 *	143 ± 4.5 *
	STI + VC	176 ± 20 ^#^	143 ± 5.0 ^#^	138 ± 5.0 ^#^	180 ± 10 ^#^	230 ± 10 ^#^

*: Represents the significant difference compared to the control group (*p* < 0.05, *n* = 10). ^#^: Represents the significant difference compared to the STI group (*p* < 0.05, *n* = 10).

**Table 7 foods-12-01691-t007:** Change of TPS activity in serum and pancreas.

	Treatment	1 Wk	2 Wk	3 Wk	4 Wk	5 Wk
Serum (U/mL)	Control	84 ± 3	92 ± 3	95 ± 3	97 ± 1	98 ± 1
	STI	44 ± 3 *	39 ± 2 *	25 ± 2 *	35 ± 3 *	44 ± 2 *
	STI + VC	68 ± 5 ^#^	62 ± 3 ^#^	48 ± 5 ^#^	63 ± 2 ^#^	69 ± 1 ^#^
Pancreas (U/mg prot)	Control	7.0 ± 0.1	7.8 ± 0.2	8.5 ± 0.3	9.1 ± 0.1	9.0 ± 0.2
	STI	3.5 ± 0.2 *	3.3 ± 0.2 *	2.1 ± 0.1 *	3.1 ± 0.1 *	3.6 ± 0.2 *
	STI + VC	5.2 ± 0.3 ^#^	4.3 ± 0.1 ^#^	3.7 ± 0.1 ^#^	4.3 ± 0.1 ^#^	5.2 ± 0.1 ^#^

*: Represents the significant difference compared to the control group (*p* < 0.05, *n* = 10). ^#^: Represents the significant difference compared to the STI group (*p* < 0.05, *n* = 10).

**Table 8 foods-12-01691-t008:** Change of SST levels in serum and pancreas.

	Treatment	1 Wk	2 Wk	3 Wk	4 Wk	5 Wk
Serum (pg/mL)	Control	50 ± 10	20 ± 5.7	20 ± 10	45 ± 4.3	50 ± 10
	STI	27 ± 3.5 *	16 ± 4.2	14 ± 0.3	19 ± 3.7 *	23 ± 2.8 *
	STI + VC	30 ± 10 *	21 ± 5.1	16 ± 5.6	30 ± 10 *	30 ± 10 *
Pancreas (pg/mg prot)	Control	29 ± 1.2	25 ± 1.5	12 ± 0.4	29 ± 2.0	98 ± 1.7
	STI	21 ± 1.8 *	19 ± 1.7 *	10 ± 0.2	20 ± 1.5 *	38 ± 0.5 *
	STI + VC	24 ± 2.3 ^#^	22 ± 0.1 ^#^	10 ± 0.1	22 ± 1.1 *	87 ± 1.6 ^#^

*: Represents the significant difference compared to the control group (*p* < 0.05, *n* = 10). ^#^: Represents the significant difference compared to the STI group (*p* < 0.05, *n* = 10).

## Data Availability

The data presented in this study are available on request from the corresponding author.

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
