# Peer review of "Effects of Soybean Trypsin Inhibitor on Pancreatic Oxidative Damage of Mice at Different Growth Periods"

_foods, 2023, doi:10.3390/foods12081691_

Round 1
Reviewer 1 Report
In this manuscript, the authors discussed the effects of the addition of a soybean trypsin inhibitor alone or combined with vitamin C in the mice's diet on pancreatic oxidative damage at different growth periods. The idea of the study was not clearly explained, nor was its benefit. Also, the introduction is not sufficient, and the conclusion section does not clarify the strength of the study. Several drawbacks in the study were listed as follows:
1. In the abstract
The main problem that the research resolved didn’t appear in the abstract section.
2. In the introduction, some information is missing in this section, such as why the author used the STI. What are its benefits for humans? Or why do they use it?
2. In the methods,
2.1. The chemical and reagents section was missing. Where did the author purchase the STI?
2.2. The animal approval number is missing.
2.3. On what basis does the author choose the STI and the Vitamin C doses? Are there any references to these doses?
2.4. The control diet also contains soybean oil (50 g/kg); does it also have TI?
2.5. How much STI is in 50 g/kg soybean oil?
2.6. The animal experiment method is not clearly described, as the author started with 10 mice in each group and then sacrificed animals every week for 5 weeks. How many mice were sacrificed every week? Did two mice generate enough data to make a difference? The author must clarify and explain.
2.7. The author claimed that they cut the mice’s pancreas into 50 to 100 mg portions; however, the mouse pancreas weighs ~ 20 mg! The author must explain.
2.8. Several sentences’ structures were wrong; please extensively revise the methodology section.
2.9. The unpublished file is repeated in the manuscript; I don’t understand why it is attached.
2.10. The primer designed for the RT-PCR for the SST, SSTR5, and TPS needs revision.
2.10. In the statistical analysis “Data were reported as mean ± SD, n = 10”, as the total group number or every week?
3. In the results
3.1. In the tables, the author claimed, “Different lowercase letters indicate a significant difference”, compared to what? Every lowercase letter must be explained.
3.2. The number of mice in each group must be written in the tables.
3.3. The author claimed, “GSH-Px activities of the three groups in both serum and pancreas decreased and then increased, and in the 3rd week reached a minimum.” Where is the comparison? Compared to what group?
34. In line number 216, “Whereas the TPS activity was significantly higher (p<0.05) than that..” The author must write the exact p-value, not just <0.05.
3.5. The full name of the SSTR5 must be written first, then the abbreviation is used.
3.6. All p values of the two compared groups must be added to the text.
3.7. The figure resolution is bad.
3.8. In the conclusion, the authors write “Supplement with VC in a soybean trypsin inhibitor diet significantly reduced the extent of oxidative damage caused by STI to the body, which further demonstrated that the harmful effect of STI on the pancreas might attribute to inducing the generation of free radicals”, It is obvious that VC can remove oxidative stress from any source. Also, STI was confirmed to induce pancreatic damage as listed in the old reference in that manuscript: “Flavin, D. F. "The effects of soybean trypsin inhibitors on the pancreas of animals and man: A review." Veterinary and human toxicology 24 (1982): 25-28”. So, what is the benefit of that study for humans or animals? Moreover, the improvement in antioxidant enzymes as well as in gene expressions was slight compared to the STI group.
Author Response
Reviewer 1:
In this manuscript, the authors discussed the effects of the addition of a soybean trypsin inhibitor alone or combined with vitamin C in the mice's diet on pancreatic oxidative damage at different growth periods. The idea of the study was not clearly explained, nor was its benefit. Also, the introduction is not sufficient, and the conclusion section does not clarify the strength of the study. Several drawbacks in the study were listed as follows:
- In the abstract
Q1: The main problem that the research resolved didn’t appear in the abstract section.
A1: Thank you for your reminder. In the revised manuscript, we have added the main problem that this study wants to resolve: “To investigate the effect of STI intake on pancreatic injury and its mechanism of action…” (Line 16-19 for details).
Q2: In the introduction, some information is missing in this section, such as why the author used the STI. What are its benefits for humans? Or why do they use it?
A2: Thank you for your reminder. In the revised manuscript, we have added the reasons why STI was chosen and its effects on the organism: “While STIs were initially developed as drugs…” (Line 64-68 for details).
- In the methods,
Q1. The chemical and reagents section were missing. Where did the author purchase the STI?
A1: Thank you for your reminder. We are very sorry that we miss the Main Reagents parts in the last manuscript. In the new manuscript, we have added part 2.1 for Main Reagents (Line 75-79 for details). The resulting serial number adjustments were also calibrated at the same time
Q2. The animal approval number is missing.
A2: Thank you for your suggestions. According to the requirement of MDPI, the animal approval number should be in the part of “Institutional Review Board Statement” (Line 420-422), which was already included in the previous version of the manuscript. But we have accepted your suggestion to avoid controversy. We had also added animal approval numbers in 2.2 (Line 92).
Q3. On what basis does the author choose the STI and the Vitamin C doses? Are there any references to these doses?
A3: Thank you for your suggestions. In determining the STI intake, we reviewed some references (Such as: Pacheco, W. J., Stark, C. R., Ferket, P. R., & Brake, J. (2014). Effects of trypsin inhibitor and particle size of expeller-extracted soybean meal on broiler live performance and weight of gizzard and pancreas. Poultry Science, 93(9), 2245-2252.), but since STI is an enzyme, its effect is heavily influenced by the enzyme activity. Since the STI in this study was prepared by our team in advance, the enzyme activity was relatively low, so the same intake dose was not seen in other studies identically. However, it was verified that this dose was beneficial for the conduct of the experiment.
Regarding the determination of VC intake, although there are no studies similar to the present study, we refer to some other references (such as: Betancor, M. B., Caballero, M. J., Terova, G., Corà, S., Saleh, R., Benitez-Santana, T., ... & Izquierdo, M. (2012). Vitamin C enhances vitamin E status and reduces oxidative stress indicators in sea bass larvae fed high DHA microdiets. Lipids, 47, 1193-1207; Roosta, Z., Hajimoradloo, A., Ghorbani, R., & Hoseinifar, S. H. (2014). The effects of dietary vitamin C on mucosal immune responses and growth performance in Caspian roach (Rutilus rutilus caspicus) fry. Fish physiology and biochemistry, 40, 1601-1607.) and generally agree that when the intake is greater than 1000 mg/kg, its intake produces a significant therapeutic effect of oxidative damage. Therefore, we conducted a pre-experiment and found that 1500 mg/kg was an intake level that could significantly produce a significant difference under the conditions of the present study. Therefore confirmed.
Q4. The control diet also contains soybean oil (50 g/kg); does it also have STI?
A4: Thank you for your detailed advice. STI is a water-soluble substance, and we determined in this study that soybean oil does not contain STI, and it was not added to soybean oil either. To avoid controversy, we had marked in Table 1 that soybean oil does not contain STI (Line 94).
Q5. How much STI is in 50 g/kg soybean oil?
A5: Thank you for your detailed advice. This question is similar to Q4, we determined in this study that soybean oil does not contain STI.
Q6. The animal experiment method is not clearly described, as the author started with 10 mice in each group and then sacrificed animals every week for 5 weeks. How many mice were sacrificed every week? Did two mice generate enough data to make a difference? The author must clarify and explain.
A6: Thank you for your very meaningful suggestions. We apologize for the unclear expressions in the last manuscript. In fact, animals put to death each week were similarly set in separate groups, which also contains 10 animals. This method of animal experimentation ensures the reproducibility required for biological experiments. The presentation of the experimental methods has been updated in the new manuscript (Line 88-89).
Q7. The author claimed that they cut the mice’s pancreas into 50 to 100 mg portions; however, the mouse pancreas weighs ~ 20 mg! The author must explain.
A7: Thank you for your detailed advice. The subject animals for this experiment were Kunming Mice, which are larger compared to strains such as C57BL/6. In reality, the weigh of pancreas is much higher than 100mg. So, we think this part is correct without modification.
Q8. Several sentences’ structures were wrong; please extensively revise the methodology section.
A8: Thank you for your detailed advice. We were aware of the language problems in the manuscript, and at the same time the editor alerted us to them. So, we sought help from native English speakers and made extensive revisions and updates to the manuscript's linguistic presentation, as detailed in the revisions in the mew manuscript.
Q9. The unpublished file is repeated in the manuscript; I don’t understand why it is attached.
A9: Thank you for your detailed advice. We did not find repeated parts in the materials and methods, but we did find a duplicate uploaded image in the experimental results, and we think you are pointing to that section. We apologize for the oversight and have re-updated the right images (Figure 2, Line 296).
Q10. The primer designed for the RT-PCR for the SST, SSTR5, and TPS needs revision.
A10: Thank you for your very meaningful suggestions. We apologize for the wrong update of Table 2. We uploaded a form from a previous study to simplify the tabulation process at the time, but forgot to change it later. Proper uploading of all primer designs has been done in the new manuscript (Line 197).
Q11. In the statistical analysis “Data were reported as mean ± SD, n = 10”, as the total group number or every week?
A11: Thank you for your detailed advice. Similar to Q6, trials were conducted weekly with 10 intra-group selections. Relevant content had been added to avoid disputes (Line 199).
- In the results
Q1. In the tables, the author claimed, “Different lowercase letters indicate a significant difference”, compared to what? Every lowercase letter must be explained.
A1: Thank you for your detailed advice. This expression refers to the difference in correlation between different groups within the same week, which is in accordance with normal writing norms. But we take your suggestion and to avoid controversy, we add " Different lowercase letters in the same column indicate a significant difference" (Below each figure and table).
Q2. The number of mice in each group must be written in the tables.
A2: Thank you for your detailed advice. The number of mice in each group has been replenished (Below each table).
Q3. The author claimed, “GSH-Px activities of the three groups in both serum and pancreas decreased and then increased, and in the 3rd week reached a minimum.” Where is the comparison? Compared to what group?
A3: Thank you for your very meaningful suggestions. The data here refer to the three groups compared to each other over five weeks, with the third week being the lowest value of all. We do not consider this expression to be problematic and have not changed it.
Q4. In line number 216, “Whereas the TPS activity was significantly higher (p<0.05) than that..” The author must write the exact p-value, not just <0.05.
A4: Thank you for your reminder. We have reservations about this question. The analysis of physical and chemical experiments is different from bioinformatics in that p<0.05 or 0.01 response is significant or highly significant for differences, and there is no requirement about explicit p-values in physical and chemical experiments as required by writing specifications. Regarding your request we have encountered it in bioinformatics studies and medical studies, but it is different from this one, so this issue we apply without modification.
Q5. The full name of the SSTR5 must be written first, then the abbreviation is used.
A5: Thank you for your detailed advice. We have added the full name of SSTR5 at the time of its first appearance in the manuscript (Line 197). Also, as it appears in the table, we have supplemented the abbreviations in the table with full names for the sake of uniformity in writing.
Q6. All p values of the two compared groups must be added to the text.
A6: Thank you for your detailed advice. This issue is similar to Q4 and we are requesting no changes.
Q7. The figure resolution is bad.
A7: Thank you for your meaningful suggestions. All figures have been redrawn (Line 274 and 297).
Q8. In the conclusion, the authors write “Supplement with VC in a soybean trypsin inhibitor diet significantly reduced the extent of oxidative damage caused by STI to the body, which further demonstrated that the harmful effect of STI on the pancreas might attribute to inducing the generation of free radicals”, It is obvious that VC can remove oxidative stress from any source. Also, STI was confirmed to induce pancreatic damage as listed in the old reference in that manuscript: “Flavin, D. F. "The effects of soybean trypsin inhibitors on the pancreas of animals and man: A review." Veterinary and human toxicology 24 (1982): 25-28”. So, what is the benefit of that study for humans or animals? Moreover, the improvement in antioxidant enzymes as well as in gene expressions was slight compared to the STI group.
A8: Thank you for your meaningful suggestions. In the conclusion of the new manuscript, we add the benefits of this study for humans and animals, as detailed in Line 402-407.
Reviewer 2 Report
The bioactive components in soybean have long been recognized for their health-promoting properties, however, the soybean trypsin inhibitor (STI) has been shown to cause metabolic disorders. This study aimed to investigate the effects of STI on the oxidative damage of the pancreas in Kunming (KM) mice during different growth periods. Results showed that STI intake caused irreversible damage to the pancreas, as evidenced by histological analysis. In addition, STI intake induced oxidative stress in the pancreas, as evidenced by the significant increase in malondialdehyde (MDA) levels in the pancreatic mitochondria of the STI group. Meanwhile, the levels of antioxidant enzymes superoxide dismutase (SOD), glutathione peroxidase (GSH-Px), trypsin (TPS), and somatostatin (SST) were decreased compared to the control group. The results of this study suggest that STI intake could lead to oxidative structural damage and pancreatic dysfunction, which may worsen over time. The findings emphasize the importance of controlling the intake of STI, especially for those who consume soybean products regularly.
Why is the intake of soybean trypsin inhibitor (STI) a concern for metabolic disorders?
Why did the STI intake cause irreversible damage to the pancreas?
Why were the levels of antioxidant enzymes decreased in the STI group compared to the control group?
Author Response
Reviewer 2:
The bioactive components in soybean have long been recognized for their health-promoting properties, however, the soybean trypsin inhibitor (STI) has been shown to cause metabolic disorders. This study aimed to investigate the effects of STI on the oxidative damage of the pancreas in Kunming (KM) mice during different growth periods. Results showed that STI intake caused irreversible damage to the pancreas, as evidenced by histological analysis. In addition, STI intake induced oxidative stress in the pancreas, as evidenced by the significant increase in malondialdehyde (MDA) levels in the pancreatic mitochondria of the STI group. Meanwhile, the levels of antioxidant enzymes superoxide dismutase (SOD), glutathione peroxidase (GSH-Px), trypsin (TPS), and somatostatin (SST) were decreased compared to the control group. The results of this study suggest that STI intake could lead to oxidative structural damage and pancreatic dysfunction, which may worsen over time. The findings emphasize the importance of controlling the intake of STI, especially for those who consume soybean products regularly.
Q1: Why is the intake of soybean trypsin inhibitor (STI) a concern for metabolic disorders?
A1: Thank you for your question. Due to the structure of the manuscript, we do not expand on this issue, in brief: when humans consume soy, the trypsin inhibitor affects the action of insulin and other digestive enzymes secreted by the human pancreas, thus affecting the body's metabolism. High intake of trypsin inhibitors may lead to reduced intestinal absorption and utilization of protein, affecting the body's absorption and utilization of nutrients, which may cause nutritional imbalance and metabolic disorders in severe cases.
Q2: Why did the STI intake cause irreversible damage to the pancreas?
A2: Thank you for your question. In this study, the reason for our conclusion is that the state of the tissue sections of the pancreas deteriorated over time without improvement due to STI intake during the experiment. The fundamental reason for this is that the trypsin inhibitor in soy can inhibit the activity of trypsin in the body, making it impossible to fully digest and absorb the protein in food, thus increasing the burden on the pancreas to secrete trypsin. This burden can accumulate to a certain level and cause tissue lesions. This is consistent with the results of other studies.
Q3: Why were the levels of antioxidant enzymes decreased in the STI group compared to the control group?
A3: Thank you for your question. In this study, the levels of antioxidant enzyme should represent whether the organ is in a normal state or not. That is, the closer the antioxidant enzyme level is to the control group, the more normal the organ level is. The abnormal pancreatic function caused by STI intake resulted in a significant decrease in antioxidant enzyme levels; due to the Vc intake, the pancreas was less damaged, so the antioxidant enzyme levels also appeared to be elevated, but were still somewhat different from the control group.
Round 2
Reviewer 1 Report
In this manuscript, the authors discussed the effects of the addition of a soybean trypsin inhibitor alone or combined with vitamin C in the mice’s diet on pancreatic oxidative damage at different growth periods. Most comments have been clarified; however, a few points still need clarification:
1. In the methodology section
1.1. The reason for choosing both STI and vitamin C doses didn’t appear in the methods.
1.2. How can mice's pancreas be cut into 50 to 100 mg portions, since it weighs ~ 20 mg! The author must explain.
2. In the results
2.1. In the tables legend “Different lowercase letters in the same column indicate a significant difference”, compared to what group? There must be an explanation for every lowercase letter. Is it compared to a control group or an STI group? Explain what each letter means.
2.2. The SD in all tables needs revision.
2.3. The authors must write the exact p-value, not just <0.05, in the whole results.
Author Response
Reviewer 1:
In this manuscript, the authors discussed the effects of the addition of a soybean trypsin inhibitor alone or combined with vitamin C in the mice’s diet on pancreatic oxidative damage at different growth periods. Most comments have been clarified; however, a few points still need clarification:
- In the methodology section
Q1: The reason for choosing both STI and vitamin C doses didn’t appear in the methods.
A1: Thank you for your reminder. In the new manuscript, we added the reason for choosing STI and VC in part 2.1, Line 76-79 for details:
“Soybean trypsin inhibitors (STI), the most common enzyme inhibitors in soybean, are the most significantly damaged to the pancreas by their ingestion. And as a strong antioxidant, Vitamin C (VC) has a theoretical potential to mitigate oxidative damage to the pancreas. Therefore, both were chosen to conduct this study.”
Q2: How can mice's pancreas be cut into 50 to 100 mg portions, since it weighs-20 mg! The author must explain.
A2: Thank you for your suggestion. We are very sorry that we did not notice our expression errors in the last version. We want to express it as 50-100 mg/100g body weight. The current version has been corrected in Lines 118-119. In addition, Kunming mice have a body weight of 35-55 g, and their pancreatic mass is also close to 100 mg, especially when volume expansion by oxidative damage to the pancreas occurs.
- In the results
Q1. In the tables legend “Different lowercase letters in the same column indicate a significant difference”, compared to what group? There must be an explanation for every lowercase letter. Is it compared to a control group or an STI group? Explain what each letter means.
A1: Thank you for your reminder. We apologize that the existing formulation is misleading. To prevent misinterpretation, we had changed the expression as recommended by you. For details under each table.
Q2. The SD in all tables needs revision.
A2: Thank you for your suggestion. We are uncertain that the SD problem you say is not because the effective number is not uniform, and we have corrected the wrong effective number digits. We accept to continue with the revision if there are additional questions.
Q3. The authors must write the exact p-value, not just <0.05, in the whole results.
A3: Thank you for your reminder. We would like to apply to you again not to make any modifications to this issue. We wrote in the experimental method that p < 0.05 would be considered significant and did not continue to delve deeper into the significance of the p-value once the difference existed. In addition, to ensure the manuscript line is coherent, the contrasts in this study were all multi-group versus multi-group comparisons, which were marked with a large difficulty in annotating each p-value. Meanwhile, there are many cases where p-values are not annotated in Foods. So we applied again without making changes.
Reviewer 2 Report
The grammar and vocabulary used in the author's responses are correct and appropriate. The responses are also clear, concise, and specific in addressing the questions asked. The author provides sufficient information to answer each question and cites relevant evidence to support their conclusions. Overall, the author's responses demonstrate a good understanding of the subject matter and their ability to communicate effectively in written English.